# OneSAM: Modality-agnostic for segment anything model in medical images

Khanh-Binh Nguyen[1][0000−0002−9948−1400]
and Chae Jung Park[1,2][0000−0002−1261−307X]

[1] National Cancer Center, Goyang-si, Gyeonggi-do, South Korea
[2] Corresponding author: cjp@ncc.re.kr

**Abstract.** Conventional deep learning segmentation models necessitate the creation of network structures and loss functions tailored to various tasks, resulting in the training of specific models. This process often leads to a considerable amount of redundant work. The Segment Anything Model (SAM) offers a comprehensive framework for managing segmentation tasks. However, the existing SAM model is primarily suitable for natural images and may demand significant computational resources during inference, creating hurdles for broad clinical adoption. In this study, we employ the adapter method to construct a modality-agnostic framework structure. During the inference stage, we use the ensemble of the baseline model with our modified model to enhance inference performance. The suggested method attains an average DSC of 0.8578 and an average NSD of 0.8679 on the validation set.

**Keywords:** Segment Anything Model · Adapter · high quality masks · Efficient segmentation learning · Cancer.

## 1 Introduction

### 1.1 Background

Many studies have applied the out-of-the-box SAM models to typical medical image segmentation tasks [6,2,1] and other challenging scenarios. For example, the concurrent studies conducted a comprehensive assessment of SAM across a diverse array of medical images, underscoring that SAM achieved satisfactory segmentation outcomes primarily on targets characterized by distinct boundaries. However, the model exhibited substantial limitations in segmenting typical medical targets with weak boundaries or low contrast. In congruence with these observations, MedSAM, a refined foundation model that significantly enhances the segmentation performance of SAM on medical images was introduced. MedSAM accomplishes this by fine-tuning SAM on an unprecedented dataset with more than one million medical image-mask pairs. Nevertheless, they still need to run on high-resource GPUs, which is not a user-friendly interaction, making advanced segmentation tools less accessible to a wider range of healthcare providers.

To realize the real-time inference speed on Laptop devices for MedSAM, in this work we proposed our OneSAM, which enhances the performance of conventional LiteMedSAM while achieving low latency. Specifically, we empirically present four simple and effective techniques to alleviate the potential performance degradation as follows:

- We adopt the SAM-HQ mask decoder for higher-quality segmentation masks.
- We design an adapter to efficiently enhance the features of the image encoder.
- We enhance the augmentation and sampling method to leverage all category masks.
- We employ the ensemble strategy to boost the performance category-by-category.
- OneSAM performs exceptionally well on both bounding box and scribble tasks.

## 1.2   Related work

The Segment Anything Model (SAM) [4] has emerged as a significant development in the field of image segmentation. Originally designed for general image segmentation, SAM has shown impressive results across various natural image segmentation tasks. However, the application of SAM in the domain of medical imaging presents unique challenges due to the complex modalities, fine anatomical structures, uncertain and complex object boundaries, and wide-range object scales present in medical images.

A notable advancement in this area is MedSAM [6], a foundation model designed to bridge the gap between general and medical image segmentation. Developed on a large-scale medical image dataset with 1,570,263 image-mask pairs, MedSAM covers 10 imaging modalities and over 30 cancer types. The model has demonstrated better accuracy and robustness than modality-wise specialist models across a wide spectrum of tasks.

Despite these advancements, there are still areas where SAM shows limitations. For instance, while SAM showed remarkable performance in some specific objects, it was found to be unstable, imperfect, or even totally failed in other situations [3]. Additionally, SAM was found to be sensitive to the randomness in the center point and tight box prompts, which could lead to a serious performance drop.

Moreover, while SAM performed better than interactive methods with one or a few points, it was outpaced as the number of points increased. These findings highlight the need for further research and development to fully harness the potential of SAM in medical image segmentation. Motivated by these downsides, in this work, we introduce OneSAM, a SAM-based method that works significantly well on medical data while keeping the inference latency low.

## 2   Method

The pipeline of the proposed efficient segmentation framework is depicted in Fig 1. The whole pipeline is based on LiteMedSAM, a distilled baseline model

from MedSAM provided by competition organizers. A detailed description of the method is as follows.

## 2.1 Preprocessing

The proposed method includes the following preprocessing as we primarily follow the baseline data processing approach for three-channel 2D data steps:

- Extract all segmentation masks of one case from ground truth images. Thus increasing the number of training cases instead of randomly choosing one category mask per training case.
- Remove all objects with less than 100 pixels and save all data in NPY format for faster training.
- Intensity normalization: First, the image is clipped to the range [-500, 500]. Then a z-score normalization is applied based on the mean and standard deviation of the intensity values.
- Resize the longest edge to 256 while maintaining the aspect ratio of the image.

In the inference phase, we handle 3D data preprocessing by conducting operations on each slice within a box. We expand each slice to three channels, and then apply the same data preprocessing method as we do for 2D data. During the training phase, to prevent high disk usage, we opt to read npz files directly for training. In the case of 3D data, we read a random slice from the stack in each iteration. To make use of more slices within the same stack, we read a single 3D data multiple times throughout each training epoch.

## 2.2 Proposed Method

The proposed method is derived from LiteMedSAM, a distilled version of Med-SAM using MobileVit as an image encoder for the semantic segmentation task. Figure 1 shows our proposed OneSAM architecture. The overall architecture of the proposed method is identical to LiteMedSAM. It mainly consists of four parts: the image adapter, the Image Encoder, the Mask Decoder, and the Prompt Encoder.

**Image Encoder Adapter** Specifically, an adapter is added at the end of each transformer block of the image encoder to enhance the features. We modify the image encoder across both the channel and spatial dimensions. For the channel dimension, we first reduce the resolution of the input feature map to $C \times 1 \times 1$ using global average pooling. Then, we employ a linear layer to shrink the channel embeddings and another linear layer to expand them, keeping a compression ratio of 0.25. Next, we calculate the weights for the channel dimension using a sigmoid function and multiply them with the input feature map to produce the input for the next layer. For the spatial dimension, we halve the spatial resolution of the feature map using a convolutional layer and restore this resolution using a

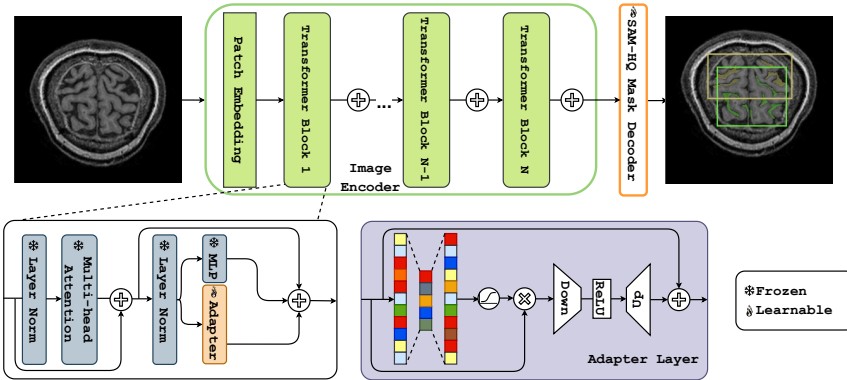

Fig. 1: Network architecture.

transposed convolution while preserving the original number of channels in the input. The overall function for the transformer block with the proposed image encoder adapter is defined as:

$$x_i^a = \text{MHSA}(\text{LN}(x_{i-1})) + x_{i-1}$$
$$x_i = \text{MLP}(\text{LN}(x_i^a)) + \text{Adapter}(\text{LN}(x_i^a)) + x_i^a \tag{1}$$

where $x_i$ and $x_i^a$ denote the output of the transformer block and multi-head self-attention (MHSA) module at the i-th layer.

**Mask decoder** We adopt the high-quality mask decoder from SAM-HQ [2], which has shown efficacy in a suite of 9 diverse segmentation datasets across different downstream tasks, where seven of them are evaluated in a zero-shot transfer protocol. However, instead of freezing the mask decoder during training, we train the whole decoder with additional parameters from the SAM-HQ mask decoder.

**Loss function** We use the summation between combinations of DiceLoss, BCELoss, and MSELoss loss because compound loss functions have been proven to be robust in various medical image segmentation tasks [5].

### 2.3   Post-processing

We ensemble two models, the baseline and our fine-tuned OneSAM for the final results. Based on the ablation study, we only select the masks from OneSAM for some categories and vice versa. We maintain the same approach as the baseline by post-processing the predicted masks through cropping and resizing to align the results with the input images.

## 3   Experiments

### 3.1   Dataset and evaluation measures

We used the challenge dataset, which is a large-scale training dataset with 1,000,000+ image-mask pairs, covering 10 medical image modalities and more than 20 cancer types. The same dataset is used for both tasks of bounding box-based and scribble-based. Throughout training, we maintain the data format as .npz. For 3D data, a random slice is read in each iteration, and during each epoch, each 3D dataset is traversed multiple times to read and train on multiple slices from the same data.

The evaluation metrics include two accuracy measures—Dice Similarity Coefficient (DSC) and Normalized Surface Dice (NSD)—alongside one efficiency measure—running time. These metrics collectively contribute to the ranking computation [7].

### 3.2   Implementation details

**Environment settings**  The development environments and requirements are presented in Table 1.

Table 1: Development environments and requirements.

| System | Ubuntu 18.04.5 LTS |
|---|---|
| CPU | Intel(R) Xeon(R) Gold 5218 CPU @ 2.30GHz (×4) |
| RAM | 252GB |
| GPU (number and type) | Two NVIDIA RTX A6000 50G |
| CUDA version | 12.0 |
| Programming language | Python 3.10 |
| Deep learning framework | torch 2.2, torchvision 0.17.1 |
| Code | https://github.com/beandkay/MedSAM-on-laptop-CVPR24 |

**Training protocols**

*1. Data augmentation*  We follow the augmentation from LiteMedSAM for the training, which resizes the longest side to 256, randomly flips along each axis, and randomly shifts the bounding box by 10px.

*2. data sampling strategy*  For the data sampling strategy, as we discussed beforehand, we extract all category masks of each image as an individual ground truth. Thus, when training, all of them will be used instead of randomly picking out only one. As a result, the number of training samples increased and improved performance.

Table 2: Training protocols.

| Pre-trained Model | LiteMedSAM [6] |
|---|---|
| Batch size | 26 |
| Patch size | 256×256×3 |
| Total epochs | 30 |
| Optimizer | AdamW |
| Initial learning rate (lr) | 1e-3 |
| Lr decay schedule | ReduceLROnPlateau |
| Training time | 340 hours |
| Loss function | DiceLoss + BCELoss + MSELoss |
| Number of model parameters | 20.218M[3] |
| Number of model trainable parameters | 20.216M[4] |
| Number of flops | 7.23G[5] |
| $CO_2$eq | 16.7 Kg[6] |

*3. Training procedure* We maintain the same optimization strategy to train the model. Specifically, the batch size is set as 24 with two NVIDIA RTX A6000 GPUs on distributed training. We use the AdamW optimizer for training, where the initial base learning rate is set as 001. We use the cosine scheduling ReduceL-ROnPlateau to decay the learning rate during the training process. The model is trained for 30 epochs.

# 4   Results and discussion

Table 3: Quantitative evaluation results bounding box task.

| Target | Baseline | | + adapter | | + SAM-HQ mask decoder | | OneSAM (Proposed) | |
|---|---|---|---|---|---|---|---|---|
| | DSC(%) | NSD(%) | DSC(%) | NSD(%) | DSC(%) | NSD (%) | DSC(%) | NSD (%) |
| CT | 81.99 | 83.69 | 87.97 | 90.61 | 89.92 | 91.84 | 89.92 | 91.84 |
| MR | 80.56 | 83.07 | 79.72 | 83.15 | 79.84 | 83.03 | 83.28 | 86.10 |
| PET | 55.10 | 29.12 | 68.85 | 54.79 | 65.92 | 51.94 | 65.92 | 51.94 |
| US | 94.77 | 96.81 | 86.01 | 90.79 | 83.18 | 88.09 | 94.78 | 96.81 |
| X-Ray | 75.82 | 80.38 | 71.10 | 76.93 | 78.27 | 84.29 | 75.83 | 80.39 |
| Dermatology | 92.47 | 93.86 | 92.74 | 94.14 | 93.54 | 95.07 | 93.54 | 95.07 |
| Endoscopy | 96.04 | 98.11 | 95.48 | 97.91 | 93.71 | 96.43 | 96.04 | 98.11 |
| Fundus | 94.81 | 96.41 | 93.02 | 94.69 | 94.60 | 96.27 | 94.81 | 96.41 |
| Microscopy | 61.63 | 65.39 | 68.55 | 75.32 | 77.92 | 84.42 | 77.92 | 84.42 |
| Average | 81.47 | 80.76 | 82.60 | 84.26 | 84.10 | 85.71 | 85.78 | 86.79 |

Table 4: Quantitative evaluation results scribble task.

| Target | Baseline | | OneSAM (Proposed) | |
|---|---|---|---|---|
| | DSC(%) | NSD(%) | DSC(%) | NSD (%) |
| CT | 81 | 83 | 83 | 86 |
| MR | 70 | 77 | 70 | 77 |
| PET | 67 | 90 | 67 | 90 |
| US | 85 | 88 | 85 | 88 |
| X-Ray | 22 | 19 | 35 | 36 |
| Dermatology | 90 | 91 | 90 | 91 |
| Endoscopy | 94 | 97 | 94 | 97 |
| Fundus | 5 | 0 | 5 | 0 |
| Microscopy | 12 | 9 | 20 | 20 |
| Average | 58 | 62 | 61 | 65 |

## 4.1 Quantitative results on validation set

The quantitative result is illustrated in Table 3 and 4 for bounding box and scribble tasks, separately. It can be found that the proposed method can achieve very promising results on low-resolution or dense object cases such as CT, PET, X-ray, Dermatology, and Microscopy. However, while improving the performance in those cases, our OneSAM performance drops in the rest of the categories compared with the baseline. We assume this could be the result of inconsistency in data pre-processing. Therefore, to get the best of every category, we ensemble the results from both models. Specifically, since OneSAM performs better on CT, PET, X-ray, Dermatology, and Microscopy but slower than the baseline, we only run the inference of OneSAM on that category and run the baseline model for the rest. Therefore, we could improve the performance and save the inference time.

Table 5: Quantitative evaluation of segmentation efficiency in terms of running time (s) for bounding box task.

| Case ID | Size | Num. Objects | Baseline | OneSAM (w/o ens) | OneSAM (w/ ens) |
|---|---|---|---|---|---|
| 3DBox_CT_0566 | (287, 512, 512) | 6 | 376.4 | 468.7 | 483.4 |
| 3DBox_CT_0888 | (237, 512, 512) | 6 | 100.5 | 123.3 | 115.3 |
| 3DBox_CT_0860 | (246, 512, 512) | 1 | 17.7 | 16.2 | 15.3 |
| 3DBox_MR_0621 | (115, 400, 400) | 6 | 157.1 | 201.1 | 147.5 |
| 3DBox_MR_0121 | (64, 290, 320) | 6 | 99.9 | 116.8 | 91.7 |
| 3DBox_MR_0179 | (84, 512, 512) | 1 | 17.1 | 15.3 | 11.9 |
| 3DBox_PET_0001 | (264, 200, 200) | 1 | 12.1 | 8.9 | 9.1 |
| 2DBox_US_0525 | (256, 256, 3) | 1 | 6.3 | 1.0 | 0.73 |
| 2DBox_X-Ray_0053 | (320, 640, 3) | 34 | 7.3 | 2.2 | 4.8 |
| 2DBox_Dermoscopy_0003 | (3024, 4032, 3) | 1 | 6.5 | 1.3 | 1.3 |
| 2DBox_Endoscopy_0086 | (480, 560, 3) | 1 | 6.1 | 1.0 | 0.7 |
| 2DBox_Fundus_0003 | (2048, 2048, 3) | 1 | 6.1 | 1.1 | 0.9 |
| 2DBox_Microscope_0008 | (1536, 2040, 3) | 19 | 6.8 | 3.2 | 3.4 |
| 2DBox_Microscope_0016 | (1920, 2560, 3) | 241 | 19.1 | 32.3 | 33.6 |

### 4.2   Segmentation efficiency results on validation set

Table 5 presents the runtime comparison between the proposed method and the baseline on selected validation sets. All times are measured from tests conducted on a local machine CPU. The average running time is 12.7 s per case in the inference phase, and the average used GPU memory is 2478 MB.

### 4.3   Qualitative results on training set

Fig 2 presents some easy and hard examples of training sets. It can be observed that OneSAM, achieves the best segmentation results on these datasets compared with the baseline. Furthermore, OneSAM can segment and cover some modalities that are wrongly segmented by the baseline.

### 4.4   Results on final testing set

### 4.5   Limitation and future work

More verification experiments could be performed to reduce resource consumption: 1) Lower the slices of 3D images to perform prediction. 2) Quantizing the model to accelerate the inference. 3) Training aware pruning and quantization methods may recover the performance.

## 5   Conclusion

The proposed method achieves the high generation ability for a wide range of medical images. The main challenge in this task lies in how to efficiently reduce the computation cost while still preserving the high performance. The proposed SAM method with low resource consumption achieves a significant improvement compared to the baseline method.

**Acknowledgements** We thank all the data owners for making the medical images publicly available and CodaLab [8] for hosting the challenge platform.

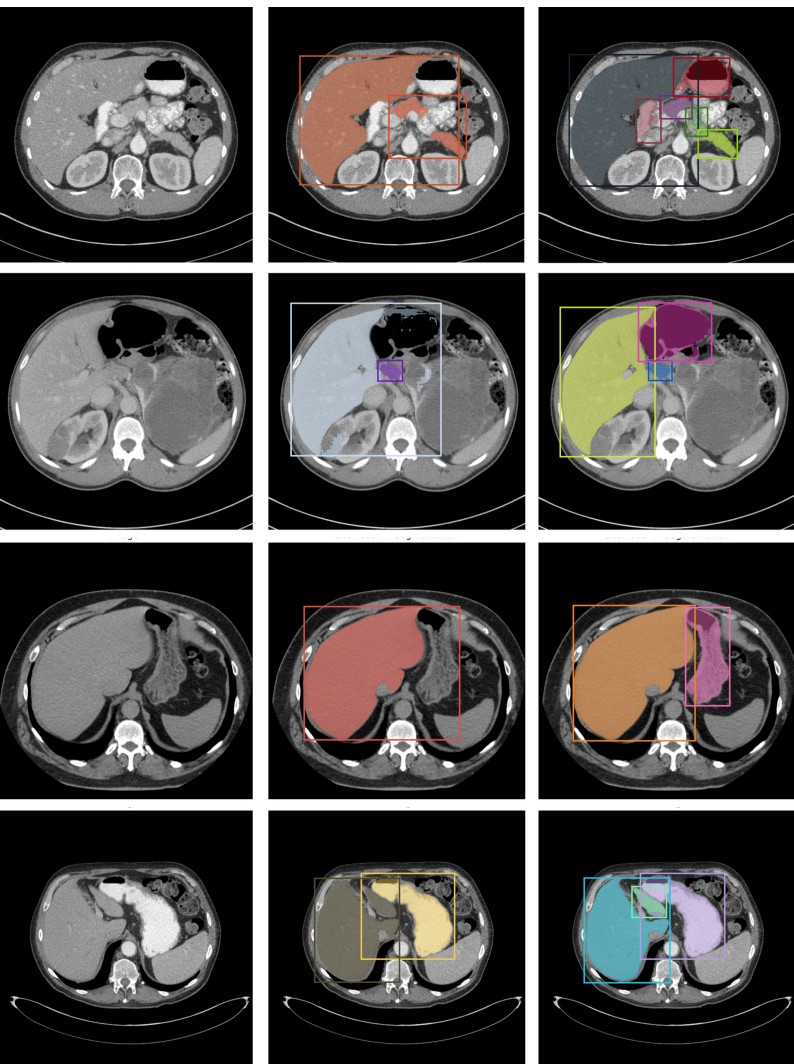

Fig. 2: Comparison between segmentation results of baseline model with proposed OneSAM. From left to right is original image, segmentation by baseline and segmentation by OneSAM.

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

Table 6: Checklist Table.

| Requirements | Answer |
|---|---|
| A meaningful title | Yes |
| The number of authors ($\leq$6) | 2 |
| Author affiliations and ORCID | Yes |
| Corresponding author email is presented | Yes |
| Validation scores are presented in the abstract | No |
| Introduction includes at least three parts: background, related work, and motivation | Yes |
| A pipeline/network figure is provided | Figure 3 |
| Pre-processing | Page 4 |
| Strategies to data augmentation | Page 6 |
| Strategies to improve model inference | Page 7 |
| Post-processing | Page 7 |
| Environment setting table is provided | Table 1 |
| Training protocol table is provided | Table 2 |
| Ablation study | Page 3 |
| Efficiency evaluation results are provided | Table 4 |
| Visualized segmentation example is provided | Figure 4 |
| Limitation and future work are presented | Yes |
| Reference format is consistent. | Yes |
| Main text $>=$ 8 pages (not include references and appendix) | Yes |

3. Ke, L., Ye, M., Danelljan, M., Tai, Y.W., Tang, C.K., Yu, F., et al.: Segment anything in high quality. Advances in Neural Information Processing Systems **36** (2024) 2

4. Kirillov, A., Mintun, E., Ravi, N., Mao, H., Rolland, C., Gustafson, L., Xiao, T., Whitehead, S., Berg, A.C., Lo, W.Y., et al.: Segment anything. In: Proceedings of the IEEE/CVF International Conference on Computer Vision. pp. 4015–4026 (2023) 2

5. Ma, J., Chen, J., Ng, M., Huang, R., Li, Y., Li, C., Yang, X., Martel, A.L.: Loss odyssey in medical image segmentation. Medical Image Analysis **71**, 102035 (2021) 4

6. Ma, J., He, Y., Li, F., Han, L., You, C., Wang, B.: Segment anything in medical images. Nature Communications **15**(1), 654 (2024) 1, 2, 6

7. Wiesenfarth, M., Reinke, A., Landman, B.A., Eisenmann, M., Saiz, L.A., Cardoso, M.J., Maier-Hein, L., Kopp-Schneider, A.: Methods and open-source toolkit for analyzing and visualizing challenge results. Scientific reports **11**(1), 2369 (2021) 5

8. Xu, Z., Escalera, S., Pavao, A., Richard, M., Tu, W.W., Yao, Q., Zhao, H., Guyon, I.: Codabench: Flexible, easy-to-use, and reproducible meta-benchmark platform. Patterns **3**(7) (2022) 8
