# OpenReview forum: "OneSAM: Modality-agnostic for segment anything model in medical images"
_thecvf.com/CVPR/2024/Workshop/MedSAMonLaptop — CVPR 2024 Workshop MedSAMonLaptop Withdrawn Submission_

### Official Review · Reviewer_rwSp · 2024-06-15
**Model adapters and alternate mask decoder for medical image segmentation using foundation models**

**Rating:** 5
**Confidence:** 4

**Review:**

## Summary

In **" OneSAM: One model for segment anything model in medical images on Laptop,"** the authors propose the use of a combination of models, i.e., LiteMedSAM and their proposed model (OneSAM). OneSAM differs from LiteMedSAM in using image encoder adapters and an alternative mask decoder. Specifically, they add image encoder adapters to each transformer block of LiteMedSAM. Additionally, they replace the LiteMedSAM mask decoder with the SAM-HQ mask decoder. OneSAM is then trained using the training dataset provided by the challenge and is evaluated against the baseline. LiteMedSAM or SAM-HQ are chosen for final inference, given their individual results across modalities. The best model for a specific modality is selected for final inference. Overall, I don't think the manuscript provided enough information for **reproducibility**.  Below, I give several suggestions to improve the manuscript's **completeness** and **reproducibility**.
 ## Detailed Comments
**Abstract:**

1- The abstract does not contain any information about the author's proposed method for the challenge, and it does not contain any results whatsoever.

**Introduction:**

2 – Background: The authors state "... enhances the performance of *conventional* LiteMedSAM". However, LiteMedSAM is not a conventional method. Here, the authors could provide more background information about the difference between MedSAM and LiteMedSAM and rephrase this sentence.
3 – Related work: The transition between the second and third paragraphs seems odd, as you moved from discussing SAM to MedSAM and then back to SAM. Additionally, this section lacks related work on model adapters and SAM-HQ as mask decoder, which seems more relevant if included, given that these are the approaches used by the authors.

**Method:**

4 – Preprocessing: There are missing details about input data dimensions.
5 – Proposed method: Given that, as the authors say "the proposed method is derived from LiteMedSAM", I'd expect at least a small description of LiteMedSAM so the authors could indicate where their innovation lies.
6 – Image encoder adapter: Could the authors provide what the acronyms in equation 1 stand for?

**Experiments:**

7 – Dataset and evaluation measures: The authors say, "The same dataset is used for both tasks". What are "both" tasks?
8 – Table 1: The provided code link is for a GitHub account profile.
9 – Training protocols – data sampling strategy: When the authors mention "the number of training samples increased", could they mention by how much and compared to what?
10 – Table 2: is the initial learning rate 1 or .001? Additionally, the authors did not provide a number of flops or CO2 footprint.
11 – Training protocols – Training procedure: It is unclear in the text whether you are just training the model adapters and mask decoder, which is what I had to assume. However, by inspecting Table 2, I see that the number of trainable parameters is virtually the same as the number of model parameters, which makes me think that you are not just tuning the model adapters and mask decoder. I would expect more detailed information here.

**Results and Discussion:**

12 – In the Results section, it is clear that the authors only used OneSAM for a few modalities and LiteMedSAM for others, where LiteMedSAM provided more accurate results. Then, I'd say their approach is a combination of modality-specific approaches. This should have been clearer early on, e.g., in the abstract.
13 – Is "+ SAM-HQ mask decoder" equal to LiteMedSAM with replaced mask decoder? That is, without adapters? Can you provide more detailed information in the table caption?
14 – In "Segmentation efficiency results on validation set", is "the average used GPU memory is 2478 MB", correct? I assume these results are determined under the challenge condition, i.e., edge device settings, which do not include a GPU.
15 – Figure 2 **only** shows examples where the authors' approach performed well.

Checklist table:
-	The title is unclear and confusing.
-	Validation scores are not presented in the abstract or the proposed approach.

---

### Official Review · Reviewer_ojmW · 2024-06-15
**generally good**

**Rating:** 6
**Confidence:** 3

**Review:**

The authors generally described their OneSAM step by step including the techiques they used such as SAM-HQ mask decoder, an adapter, data augmentation and so on while some details can be elaborated more. The ablation study's result is somewhat not clear, the baseline performance is lower than the one showed on the codabench. Also, the running time of 2D decreased a lot while the 3D objects running time even got longer, the author should explain possible reasons. More elaborations should be add for Qualitative result.

---

### Official Review · Reviewer_wauX · 2024-06-16
**Attempt on both image encoder and mask decoder to improve overall segmentation results, but speed is the major concern**

**Rating:** 6
**Confidence:** 5

**Review:**

Instead of using a different Image encoder, this work tried to use an adapter at the end of each transformer.

For the mask decoder, SAM-HQ from Ref[2] is used.

The methods are explained mostly in details, including preprocessing, major framework and data sampling strategy.
But SAM-HQ still needs to be explained, as this is not commonly known.
How the adapter is trained, is not explained in details either.

The major drawback of this work, is the speed, which is shown to be substantially slower even than the baseline.

 BTW, the adapter is not improving a lot from baseline shown in the ablation study.

Section 4.2, 4.3 and 4.4 are too simple, which lacks the analysis on speed, and model performance.

---

### Decision · Program_Chairs · 2024-10-01

**Decision:**

Major Revision

**Comment:**

Please address the reviewer comments add your testing results.